# A Multiscale Self-Adaptive Attention Network for Remote Sensing Scene Classification

**Lingling Li** [1] , **Pujiang Liang** [1] , **Jingjing Ma** [1] , **Licheng Jiao** [1,*] , **Xiaohui Guo** [1] , **Fang Liu** [1] **and Chen Sun** [2]

1   Key Laboratory of Intelligent Perception and Image Understanding of Ministry of Education, International Research Center for Intelligent Perception and Computation, Joint International Research Laboratory of Intelligent Perception and Computation, School of Artificial Intelligence, Xidian University, Xi'an 710071, China; llli@xidian.edu.cn (L.L.); pjliang@stu.xidian.edu.cn (P.L.); jjma@xidian.edu.cn (J.M.); xguo_18@stu.xidian.edu (X.G.); fliu@xidian.edu.cn (F.L.)
2   The Fifth Electronics Research Institute of Ministry of Industry and Information Technology, Guangzhou 510610, China; jtpei@stu.xidian.edu.cn
*   Correspondence: lchjiao@mail.xidian.edu.cn

**Abstract:** High-resolution optical remote sensing image classification is an important research direction in the field of computer vision. It is difficult to extract the rich semantic information from remote sensing images with many objects. In this paper, a multiscale self-adaptive attention network (MSAA-Net) is proposed for the optical remote sensing image classification, which includes multiscale feature extraction, adaptive information fusion, and classification. In the first part, two parallel convolution blocks with different receptive fields are adopted to capture multiscale features. Then, the squeeze process is used to obtain global information and the excitation process is used to learn the weights in different channels, which can adaptively select useful information from multiscale features. Furthermore, the high-level features are classified by many residual blocks with an attention mechanism and a fully connected layer. Experiments were conducted using the UC Merced, NWPU, and the Google SIRI-WHU datasets. Compared to the state-of-the-art methods, the MSAA-Net has great effect and robustness, with average accuracies of 94.52%, 95.01%, and 95.21% on the three widely used remote sensing datasets.

**Keywords:** multiscale feature; feature fusion; scene classification; dual attention

## 1. Introduction

With the rapid development of remote sensing technology and the increasing number of satellites, more and more solid sources of data-support for land use investigation can be obtained [1–3]. Images used in remote sensing image processing and interpretation tasks contain more complex structures and have higher resolutions than before. The traditional method based on a single pixel with physical information is no longer suitable for complex remote sensing image classification. In recent years, some methods based on the entire image content have been used to extract high-level semantic information in remote sensing images [4,5].

The method for remote image sensing classification commonly includes two parts: a features extractor and a classifier. According to the method of extracting features, the methods can generally be divided into two groups: traditional methods and deep learning (DL) methods. Traditional methods are usually based on hand-crafted features, which are used to extract low-level features including surface information such as color and texture. In contrast, DL methods can extract more robust features through convolution operations and hierarchical structures, which can obtain high-level

features including abstract information, such as objects in the image. In recent years, DL methods have gradually become mainstream in many artificial intelligence tasks such as speech recognition [6,7], semantic segmentation [8,9], and image retrieval [10].

In traditional methods, sparse representations [11,12] and morphological profiles (MPs) [13] are typically used as the fundamental feature extractor [14–16]. For example, Yin et al. [12] proposed a remote sensing image fusion algorithm with sparse representations and color matching instead of the intensity hue saturation (IHS) color model and Brovey transform. Logistic regression [17,18], decision tree [19], random forest [20], extreme learning machine (ELM), probabilistic neural network (PNN), and support vector machines (SVM) [21,22] are usually adopted as the classifiers. For example, Thaseen et al. [23] constructed a multiclass SVM to decrease the training and testing time and increase the individual classification accuracy. Han et al. [24] compared the behavior of random forest with ELM, PNN, and SVM for the intelligent diagnosis of rotating machinery, which proved that random forest outperforms the comparative classifiers in terms of recognition accuracy, stability, and robustness to features, especially with a small training set. In general, traditional methods often capture shallow information and so it is difficult for them to achieve excellent performance. Therefore, DL methods are more suitable for remote sensing image classification with rich semantic information.

As mentioned above, feature extractors and classifiers usually are separated in traditional methods. In DL methods, classification models are generally end-to-end, so feature extractors and classifiers are trained and predicted simultaneously. Due to their good performance, DL methods have also been widely used in remote sensing image processing and interpretation, like hyperspectral pixel classification [25–28] and scene classification [29,30]. In order to deal with different problems flexibly, the basic convolution operation also has many variants, such as dilated convolution [31], deformable convolution [32], and transposed convolution [33]. For example, Rizaldy et al. [34] introduced a multiscale fully connected network (MS-FCN) with dilated convolution operation and a fully connected network (FCN) [35] to minimize the loss of information during the point-to-image conversion. Körez et al. [36] proposed a multiscale Faster R-CNN method with deformable convolution for single/low graphics processing unit (GPU) systems, which can make up for the limitations of the anchor's shape. Wang et al. [33] proposed an anchor-free object detection method with a transposed convolution operation. It can be seen from the current development that the convolution operation is an excellent feature extraction method.

In image classification, many classical models have been proposed, such as AlexNet [37], GoogLeNet [38], and VGG [39]. In addition, there are many improved classification methods. He et al. [40] proposed ResNet by adding a skip connection between two specific layers, which can alleviate the gradient disappearance problem during backpropagation. Xie et al. [41] proposed ResNeXt by introducing a split–transform–merge operation in Inception [38] to ResNet. Huang et al. [42] proposed DenseNet, aiming to make the output of a block contain information for all layers in this block. DenseNet consists of many dense blocks and a classifying layer like ResNet, and the inputs of each layer are the sum of the outputs of all the previous layers in each dense block. Chen et al. [43] took advantage of both ResNet and DenseNet and proposed a Dual-Path Network (DPN), which consists of a residual block and a dense block in parallel. Hu et al. [44] introduced Squeeze-and-Excitation (SE) to ResNet and proposed SE-Net. In the SE-block, the output features are first squeezed by max-pooling and then transformed into a group of weights by using two fully connected layers, and the weights are multiplied to original features in each channel.

The above models, based on convolutional neural networks, have been used in many remote sensing tasks. In the past few years, modeling of higher-order statistics for more discriminative image representations has attracted great interest in deep ConvNets. In 2019, many new methods based on second-order features in convolutional neural networks have been proposed [45–48], and have achieved good results in many image processing tasks, such as image classification [45], semantic segmentation [46], image super-resolution [47], and pedestrian re-identification [48]. Gao et al. [45] proposed the GSoP-Net model, introducing Global Second-order Pooling (GSoP)

from lower to higher layers for exploiting holistic image information. After nonlinear transformation, a covariance matrix obtained by the GSoP layer is used for tensor scaling along the channel dimension, so that GSoP-Net can make full use of the second-order statistics of the holistic image. At the same time, the attention mechanism in convolutional neural networks has also developed in many directions in image processing tasks. Wang et al. [49], with an Efficient Channel Attention (ECA) module, proposed ECA-Net, which only involves k (<9) parameters but brings about a clear performance gain.

However, the features in the models mentioned above are all single scale, and the scale variation of the objects in image has a great influence on the models. As shown in Figure 1, it is difficult to classify remote sensing scenes if the size of the objects changes a lot. Some studies are aiming to solve this problem. In [50], the original images were cropped into different sizes and rescaled to original size, then the generated different scale images were used as inputs and a scale-aware model was acquired.

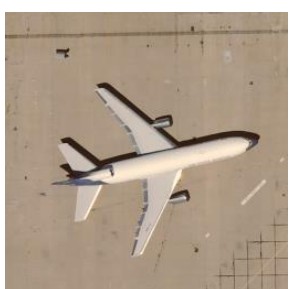 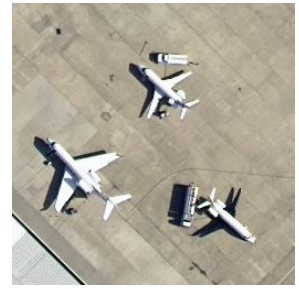 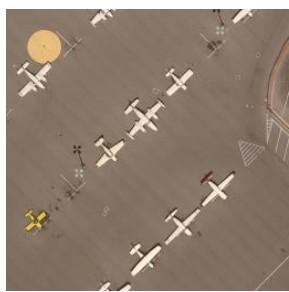

**Figure 1.** Examples of object scale variation in remote sensing images.

Zeng et al. [51] proposed a novel end-to-end CNN by integrating the global-context features (GCFs) and local object-level features (LOFs), which allows the method to be more discriminative in scene classification. Liu et al. [30] trained a CNN with multiple size samples and used the spatial pyramid pooling (SPP) [52] method to make inputs with different sizes and outputs with the same size.

The methods for scene classification mentioned above mostly have the same receptive field, not taking full advantage of the flexibility of the convolution operation. The traditional image multiscale method needs to crop or resize the inputs, which will lead to some loss of detail. In addition, if the input size of the convolutional neural network is not fixed, additional methods, such as SPP, need to be introduced in the prediction. Due to the hierarchical structure of convolutional neural networks, the fusion of features of different layers has become a new multiscale method, and has achieved good results in the field of target detection. However, shallow low-level features are not robust, so the multiscale features obtained by this method are not suitable for classification tasks. A new method is needed to solve the problem that the scale variation of the objects in the image has a great influence on the models. The dilated convolution can expand the receptive field of the convolutional kernel without introducing additional parameters, so that the convolutional kernel can focus on objects of different sizes. Therefore, using the characteristics of dilated convolution, a novel multiscale features extraction module is proposed in this paper to extract features of objects of different sizes in the image.

In addition, the methods of directly adding or concatenating the two feature maps mean that the contribution of the two feature maps to the entire model is equal. In [53], the features from different layers are combined by adding them element-wise. The local information in the shallow layer and the semantic information in the deep layer are fused, and no additional parameters are introduced. In [38], the structure of inception combines features from different kernels by concatenating in different channels, which will introduce some parameters in the next layer. In fact, these methods are often too rigid and cannot fully utilize the information contained in the two input feature maps. The channel attention mechanism can assign different weights to each channel, so as to effectively select which channel information is more important. Therefore, the channel attention mechanism with a squeeze-and-excitation operation can be used to adaptively select the importance of the

corresponding channels in the two feature maps through global information. By taking advantage of the squeeze-and-excitation operation, a multiscale feature-fusion module is proposed to fuse the two input feature maps according to the contribution to the model.

According to the above analysis of multiscale feature extraction and feature fusion methods, MSAA-Net is proposed for the problem of the object size in the same category image varying greatly. Different from the methods mentioned above using different sizes of samples as inputs, a low-level feature extraction module with different receptive fields is adopted to capture multiscale features. Then multiscale features are fed into a feature-fusion module to merge adaptively. Finally, combining the classic residual block structure and incorporating the attention mechanism, a deep feature extraction module is designed. In this module, the skipping connection in the residual block can solve the problem of gradient disappearance, and the attention mechanism can perform sparse and enhanced processing on the features extracted by convolution. Therefore, the deep classification module can enhance fusion features via a self-attention convolution block and output the final classification results.

The main contributions of this paper are as follows.

1. A novel multiscale features extraction module containing two convolution block branches is proposed to extract features at different scales. The two branches have the same structure, but different receptive fields. Convolutional kernels of different receptive field sizes can capture features of different scales, and no additional parameters are introduced, because the parameters in both branches are shared.
2. A multiscale feature-fusion module is designed for the proposed network. In this module, a squeeze process is used to obtain global information and the excitation process is used to learn the weights in different channels. With global information, the proposed method can select more useful information from the two feature maps for adaptive fusion.
3. A deep classification module with the attention mechanism is proposed to extract high-level semantic features and generate final classification results. In this module, the skipping connection can well solve the problem of gradient disappearance, and the attention mechanism can perform sparse and enhanced processing on the features.

The rest of the paper is organized as follows. Section 2 presents the proposed method in detail. Section 3 presents the data and experimental results. A discussion is provided in Section 4. Finally, the conclusions are provided in Section 5.

## 2. Materials and Methods

In this section, the proposed method, MSAA-Net, is introduced in detail. As shown in Figure 2, the proposed network contains three parts: a multiscale features extraction module, a feature-fusion module, and a deep classification module. In the multiscale features extraction module, two parallel convolutional layers with the same structure but different dilation rates are used to capture multiscale features. In the feature fusion module, a squeeze-and-excitation operation is used to adaptively fuse features with different scales. The deep classification module contains many basic convolutional blocks connected in series. Each basic block contains two convolutional layers with an attention mechanism and shortcut connection, which can obtain more robust semantic information and effectively suppress overfitting. In Figure 2, the number after the convolution block denotes the size of channels of the output. Convolutional kernels with a stride of 2 are used to reduce the size of the feature maps. The output of convolutional layers is vectorized by global average pooling, followed by a fully connected layer. In the following, the structures of the three modules are separately described in detail.

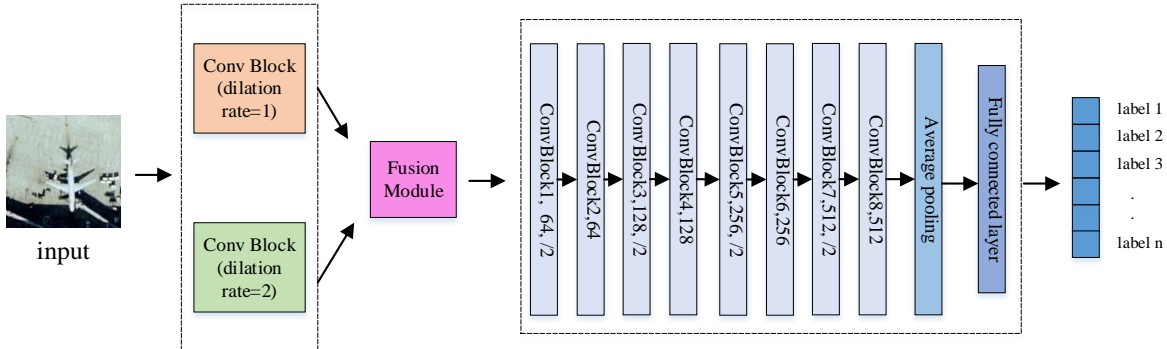

**Figure 2.** Overview of the proposed method.

## 2.1. Multiscale Features Extraction Module

In recent years, dilated convolution has been used as a multiscale extraction method in semantic segmentation [54] and object detection [55] tasks and has achieved quite good results. Dilated convolution up-samples filters by inserting zeros between weights, as illustrated in Figure 3. It enlarges the receptive field, but does not introduce extra parameters. Dilated convolution can extract multiscale features without reducing the spatial resolution of responses, which is the key difference from other multiscale extraction methods.

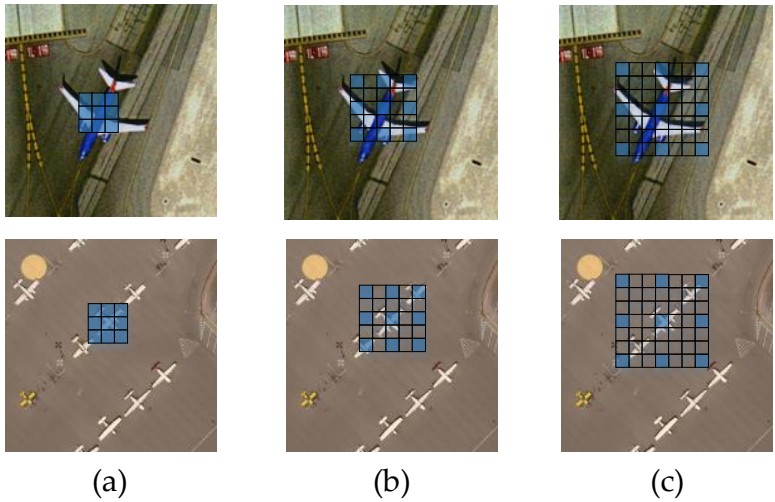

(a)        (b)        (c)

**Figure 3.** Effects of different receptive fields on different samples. (**a**) Receptive field of $3 \times 3$ convolutional kernel with dilation rate = 1. (**b**) Receptive field of $3 \times 3$ convolutional kernel with dilation rate = 2. (**c**) Receptive field of $3 \times 3$ convolutional kernel with dilation rate = 3.

First, we construct two parallel branches with the same structure but different receptive fields to extract features in different scales. In each branch, there are two convolutional layers with a kernel size of $3 \times 3$. In the convolutional layer, the size of the receptive field can be adjusted according to different dilation rates. As shown in Figure 3, receptive fields of different sizes have different feature extraction capabilities for objects of different sizes. In the one branch, the dilation rate is set to 1 with a kernel size of $3 \times 3$. In the other branch, the dilation rate is set to 2 with a kernel size of $3 \times 3$, which means that its receptive field is same as the layer with a kernel size of $5 \times 5$.

In the multiscale features extraction module, given the input $X \in R^{3 \times H \times W}$, two output feature maps $O_1$ and $O_2$ are obtained as follows:

$$O_1 = f(f(X * W_1) * W_3) \tag{1}$$

$$O_2 = f(f(X * W_2) * W_3) \tag{2}$$

where $W_1$, $W_2$, and $W_3$ represent the convolutional parameters. The values in $W_1$ and $W_2$ are equal, but the dilation rate is different. $*$ is the convolution operation and $f$ refers to the ReLU activation function. The convolutional layers in the two branches have the same stride and structure, so the size and channel of $O_1$ and $O_2$ are the same. Due to different receptive fields in each branch, there is different information about scale in $O_1$ and $O_2$, and then they are fed into the feature-fusion module.

## 2.2. Multiscale Feature-Fusion Module

Element-wise addition and concatenation in channels are currently two widely used feature fusion methods that treat the contribution of the fused features to the model as equivalent. However, because the input features contain different scales of information, their contribution to the model is often different. To address this issue, an adaptive feature fusion module is proposed. As shown in Figure 4, the two groups of features from two branches are fused by the adaptive fusion module. $O_1$ and $O_2$ denote two feature maps from the multiscale feature extraction module, which contains features in different scales. $O$ denotes the sum of $O_1$ and $O_2$, and $z$ is the global features of $O$. $p$ and $q$ denote two groups of weights for $O_1$ and $O_2$, $fc$ is the fully convolutional layer, and *softmax* represents the softmax function. The output multiscale features of this part are used as the input of the classification module.

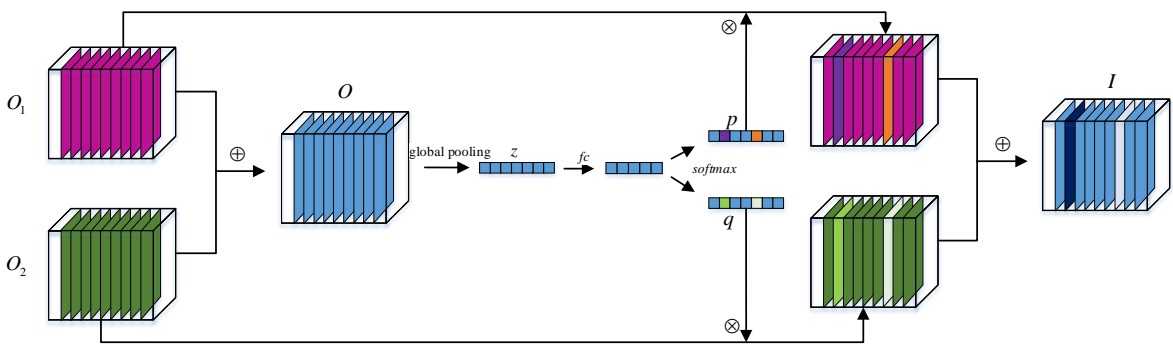

**Figure 4.** Illustration of multiscale feature-fusion module.

The squeeze-and-excitation (SE) process is a classic method to make each channel have different weights. SE is adopted to make the network adaptively select useful information based on its contribution to the classification from two input features in the proposed methods.

First, the features $O_1$ size of $C \times W \times H$ and $O_2$ size of $C \times W \times H$ are added element-wise:

$$O = O_1 + O_2 \tag{3}$$

Then the feature map $O$ size of $C \times W \times H$ is squeezed to global feature $z$ size of $C \times 1$ by using global average pooling. The *c-th* element of $z$ is obtained by shrinking the *c-th* channel of $O$ through spatial dimensions $H \times W$, so that the *c-th* of $z$ is calculated by:

$$z_c = F_{gp}(O_c) = \frac{1}{H \times W} \sum_{i=1}^{H} \sum_{j=1}^{W} O_c(i,j) \tag{4}$$

where $O_c$ denotes the *c-th* channel of $O$. Then the two fully connected layers and softmax function are used to generate two groups of weights, $p$ and $q$, by the following formulas:

$$p' = fc_2(fc_1(z)) \tag{5}$$

$$q' = fc_3(fc_1(z)) \tag{6}$$

$$\begin{bmatrix} p \\ q \end{bmatrix} = \delta \begin{bmatrix} p' \\ q' \end{bmatrix} \tag{7}$$

where $fc_2$ and $fc_3$ have the same structure but different parameters. $p'$ and $q'$ are two groups of parameters generated by the fully connected layer, and their values represent the importance of information of different channels in the two feature maps. $\delta$ means to perform softmax operation on the corresponding positions of $p'$ and $q'$, which means the sum of the corresponding position elements in $p$ and $q$ is 1 and the element in $p$ and $q$ ranges from 0 to 1. The final feature map $I$ can be obtained through the weights $p$ and $q$ in various channels:

$$I = p \cdot O_1 + q \cdot O_2 \tag{8}$$

The key point of this part is adaptively selecting effective information in two input features, which differ in scale. By using global average pooling on the input feature map for each channel, the input feature map can be compressed to one dimension, where each element contains global information. The two fully connected layers can generate different weights based on the importance of information for different channels during the training process.

The softmax function can weigh the importance of the corresponding channels in the two input feature maps, so that the corresponding channels in the two feature maps are fused at a certain ratio. Different from the general methods treating $O_1$ and $O_2$ as equally important, the proposed method gives input features weights in different channels, which can reduce redundant information and extract useful information from the input. The weights in different channels are learned by global average pooling, two fully connected layers, and a softmax function, so the weights in different channel are obtained adaptively. There are multiscale features in the output feature map that are more useful for the classification module.

### 2.3. Deep Classification Module

The attention mechanism is widely used in image processing tasks because it can extract more information that is helpful for classification and detection. Hu et al. [44] exploits the interchannel relationship by introducing a squeeze-and-excitation module. Woo et al. [56] shows that spatial attention plays an important role in deciding where to focus. In this part, a convolution block containing channel attention and spatial attention is designed for the deep classification module. This module consists of eight basic blocks, a global pooling layer, and a fully connected layer. Figure 5 shows the illustration of a basic block. Each block contains two convolutional layers, batch normalization, ReLU activation function, a channel attention module, and a spatial attention module. The final output is the sum of input and output of the spatial attention layer.

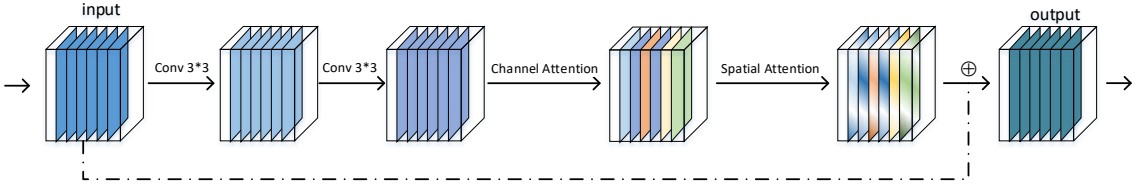

**Figure 5.** Structure of a basic convolution block with attention.

In a classical residual block, let $F(X)$ be the function to be learned by a residual block. $F(X)$ can be expanded as follows:

$$F(X) = f(f(X * W_1) * W_2) + X \tag{9}$$

where $W_1$ and $W_2$ represent the convolutional kernels of two convolutional layer, and $f$ refers to the ReLU activation function.

In the proposed basic convolution block, the function $H(X)$ from input to output can be expanded as follows:

$$H(X) = A_s(A_c(f(f(X * W_1) * W_2))) + X \tag{10}$$

where $A_s$ and $A_c$ denote channel attention and spatial attention operations. On the one hand, the multiscale information in the proposed model is distributed in different channels, so the channel attention layer can further enhance the multiscale features and make it sparse. On the other hand, high-level semantic information usually comes from the objects in the image, so the spatial attention mechanism can enhance the ability to focus on the objects. In the proposed convolutional block, the specific implementation of the channel attention mechanism and spatial attention mechanism is introduced as below.

### 2.3.1. Channel Attention

As shown in Figure 6, both average pooling and max pooling are used first to obtain global information. Then a multilayer perceptron (MLP) with two fully connected layers is set to learn different weights in channels. Finally, the sum of the two groups of weights from the average pooling feature and max pooling feature is considered as the attention in different channels. The channel attention is computed as follows:

$$\begin{aligned} A_c(Y) &= Y \otimes (\sigma(MLP(Avgpool(Y)) + MLP(Maxpool(Y)))) \\ &= Y \otimes (\sigma(W_1(W_0(Y_{avg}^c)) + W_1(W_0(Y_{\max}^c)))) \end{aligned} \tag{11}$$

where $\sigma$ denotes the sigmoid function, $W_0 \in \mathbb{R}^{C/r \times C}$ and $W_1 \in \mathbb{R}^{C \times C/r}$. $\otimes$ means multiply by the corresponding channel. For average pooling and max pooling, $W_0$ and $W_1$ are shared for both inputs. It can be seen that channel attention uses the global information of the feature map itself to generate different weights for different channels. Without the introduction of external information, channel attention can enhance useful information and reduce the impact of redundant information.

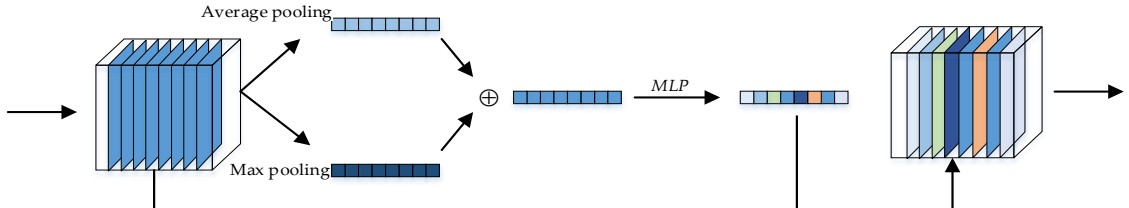

**Figure 6.** The channel attention module.

### 2.3.2. Spatial Attention

The structure of the spatial attention module is shown in Figure 7. The input is pooled along the channel axis by max pooling and average pooling. After concatenating the two sets of global information, a convolutional layer is used to generate weights for different spatial positions. Multiplying the input by the weight gives the output. Different from channel attention pooling in a channel, average pooling and max pooling in different locations are used to find where is important. The two pooling features are concatenated in channels, and a convolutional layer is used to generate different weights in location. The spatial attention is computed as follows:

$$\begin{aligned} A_s(Y) &= Y \odot \sigma(conv^{7 \times 7}([Avgpool(Y); Maxpool(Y)])) \\ &= Y \odot \sigma(conv^{7 \times 7}([Y_{avg}^s; Y_{\max}^s])) \end{aligned} \tag{12}$$

where $\sigma$ denotes the sigmoid function and $conv^{7 \times 7}$ represents a convolutional layer with a kernel size of $7 \times 7$. $\odot$ means multiplying the corresponding position. It can be seen that the spatial attention mechanism can generate different weights for each location by convolving the global information.

By multiplying the weights of the positions corresponding to the feature map, the information at important positions can be effectively enhanced and the interference of the information at unimportant positions can be reduced.

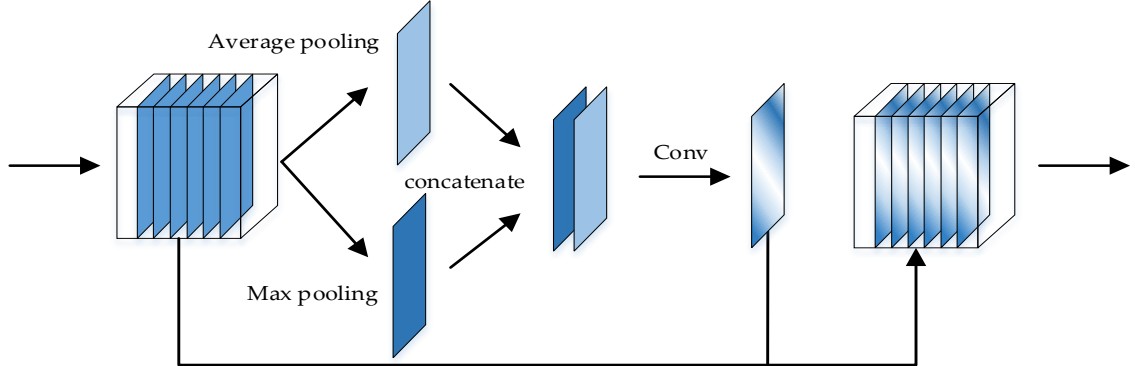

**Figure 7.** The spatial attention module.

The proposed MSAA-Net is an end-to-end network, which is easy to train and test. MSAA-Net can extract multiple features that are fused with adaptive weights based on the importance of the information. We expect that the proposed method will be good for solving the problem of the scale variation of the objects having a great influence on the model in remote sensing image classification.

## 3. Experimental Results

In order to demonstrate that the proposed method is effective, the UC Merced dataset, the NWPU dataset, and the Google dataset of SIRI-WHU are adopted in the experiments. The traditional methods of locality-constrained linear coding (LLC) [57] and bag-of-visual-words (BoVW) were compared. For deep learning methods, the classic models AlexNet [37], GoogLeNet [38], and VGG-16 [39] were compared and some excellent models, such as ECA-Net [49] and GSoP-Net [45], proposed last year, were also compared. Furthermore, the ResNet-18 is roughly the same as our network in structure except for the multiscale feature extraction module and the feature-fusion module, so ResNet-18 is used for comparison. The experiments were performed on an HP-Z840 Workstation with a single Nvidia RTX2080Ti GPU and 128-GB RAM under Ubuntu 16.04 LTS with CUDA 10.0 release, using the deep learning framework of Pytorch. The parameters in the training phase are as follows. The maximum iteration is set to 10K, and the batch size is 16. The weight decay is set to 0.0005 and the learning rate is 0.01 in the beginning, decaying by dividing by 10 every 4000 iterations.

### 3.1. UC Merced Land Use Dataset

The UC Merced land use dataset was collected from large optical images of the United States Geological Survey, containing 21 categories. There are 100 images in every class, which have a size of $256 \times 256$ pixels and a spatial resolution of one foot per pixel. Eighty percent of the dataset in each category is used as random training samples, and the rest is used as test samples. Each experiment was repeated 10 times, and the average classification accuracy was recorded. The 21 classes are agriculture, airplane, baseball diamond, beach, buildings, chaparral, dense residential, forest, freeway, golf course, harbor, intersection, medium residential, mobile home park, overpass, parking lot, river, runway, sparse residential, storage tanks, and tennis court. Figure 8 shows an image from each class.

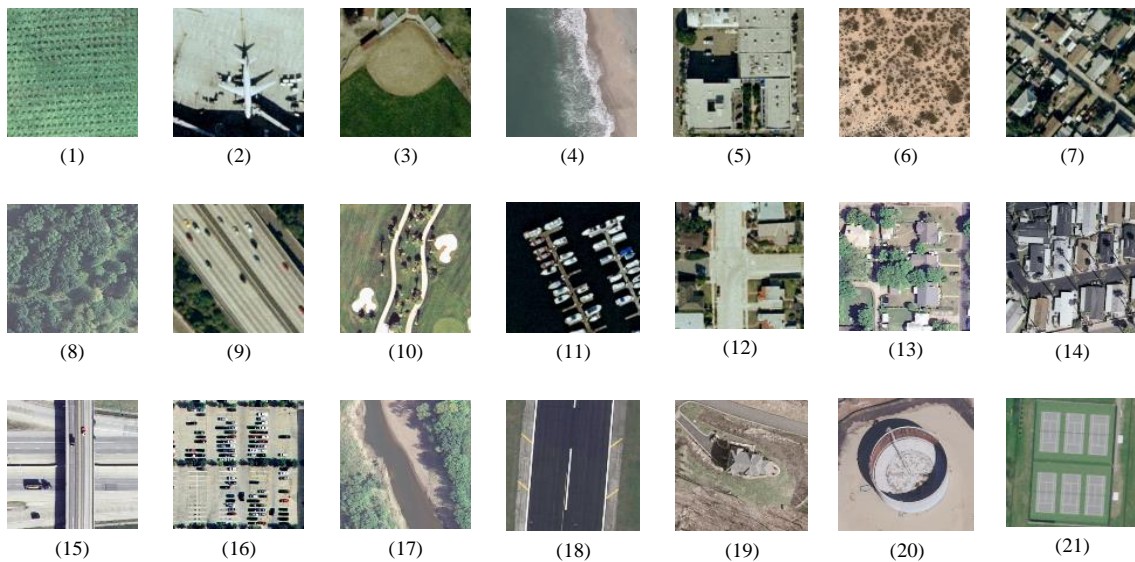

**Figure 8.** Representation of classes in the UC Merced dataset. (1) Agriculture. (2) Airplane. (3) Baseball diamond. (4) Beach. (5) Buildings. (6) Chaparral. (7) Dense residential. (8) Forest. (9) Freeway. (10) Golf course. (11) Harbor. (12) Intersection. (13) Medium residential. (14) Mobile home park. (15) Overpass. (16) Parking lot. (17) River. (18) Runway. (19) Sparse residential. (20) Storage tanks. (21) Tennis court.

As shown in Table 1, the traditional methods of LLC and BoVW are compared, and the deep learning methods AlexNet, GoogLeNet, VGG-16, ECA-Net, and GSoP-Net follow. It is obvious that the deep learning methods perform better than traditional methods based on handcrafted features, which demonstrates that deep semantic features are more useful for remote sensing scene classification. Compared with GSoP-Net, which introduces second-order features into a convolutional neural network and combines the classic structure of ResNet, the proposed MSAA-Net is 1.9% better. It can be seen that, compared with some classic networks, such as VGG and GoogLeNet, the proposed method also has better classification results. As shown in Figure 9, the proposed method achieves the best results in these models, demonstrating that MSAA-Net is effective.

**Table 1.** Comparison between the previously reported accuracies for the UC Merced dataset.

| Method | Classification Accurzacy (%) |
| --- | --- |
| LLC | 82.85 |
| BoVW | 73.46 |
| AlexNet | 85.63 ± 2.6 |
| GoogLeNet | 92.81 ± 0.64 |
| VGG-16 | 89.09 ± 2.01 |
| GSoP-Net | 92.62 ± 1.2 |
| ECA-Net | 94.05 ± 0.96 |
| ResNet-18 | 90.95 ± 0.42 |
| MSAA-Net without attention | 92.38 ± 0.35 |
| MSAA-Net (ours) | **94.524 ± 0.74** |

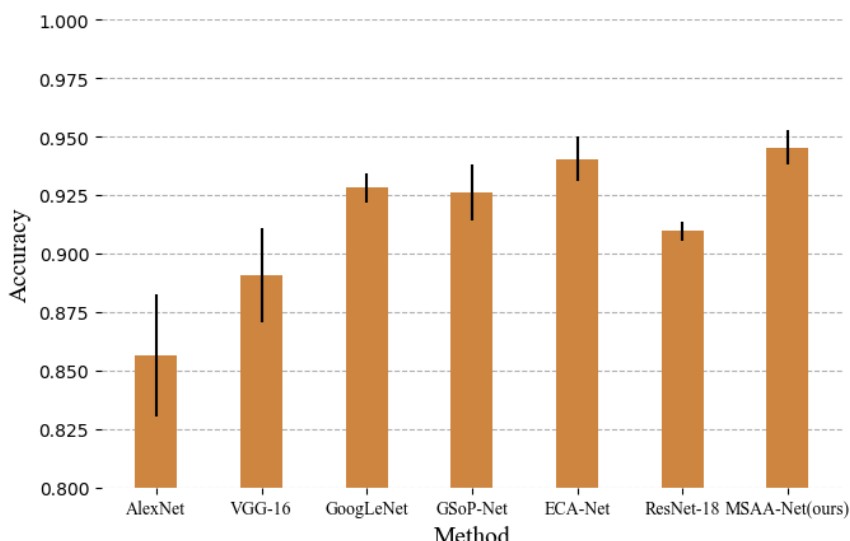

**Figure 9.** Classification results of different methods for the UC Merced dataset.

The proposed network is nearly the same as ResNet-18 except for the multiscale-feature extraction module, the feature-fusion module, and the attention module, so ResNet-18 is compared as a baseline. The network with a multiscale extraction module performs 1.4% better than without a multiscale extraction module, which demonstrates the effectiveness of the multiscale extraction module. Furthermore, the complete network performs 1.2% better than the network without an attention module. The reason for this is that the attention module can enhance information in features, which is useful for classification, and for features in MSAA-Net different in scale, the attention module can help extract a similar part in features from different scales. As shown in Figure 10, compared with ResNet-18, the proposed network achieves the highest accuracy in 21 classes, especially for the airplane and storage tank, where the object varies greatly in size.

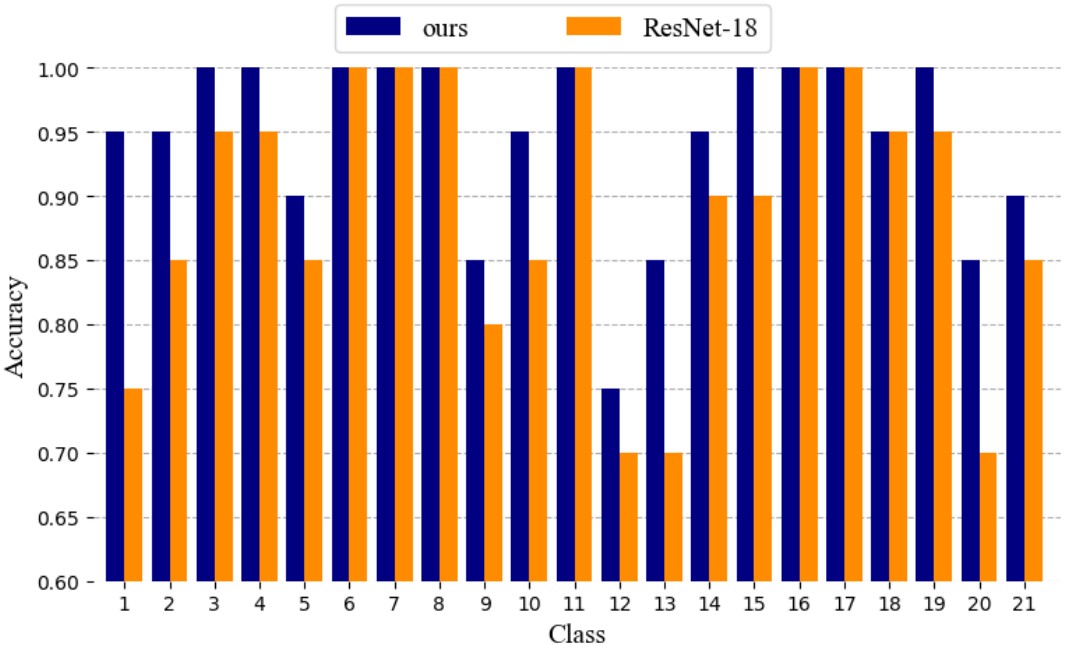

**Figure 10.** Accuracy in each class of ResNet-18 and multiscale self-adaptive attention network (MSAA-Net) for the UC Merced dataset.

### 3.2. NWPU-RESISC45 Dataset

In order to further demonstrate the effectiveness of the proposed method, the NWPU-RESISC45 [58] dataset, which was created by Northwestern Polytechnical University (NWPU), was also tested. This dataset contains 31,500 images, covering 45 scene classes with 700 images in each class. These 45 scene classes include airplane, airport, baseball diamond, basketball court, beach, bridge, chaparral, church, circular farmland, cloud, commercial area, dense residential, desert, forest, freeway, golf course, ground track field, harbor, industrial area, intersection, island, lake, meadow, medium residential, mobile home park, mountain, overpass, palace, parking lot, railway, railway station, rectangular farmland, river, round about, runway, sea ice, ship, snowberg, sparse residential, stadium, storage tank, tennis court, terrace, thermal power station, and wetland. The image size is 256 × 256 pixels, with a spatial resolution varying from 0.2 to 30 m per pixel. The ratio of the number of training samples is set to 50%, and the remainder are used for testing. Each experiment was repeated 10 times, and the average classification accuracy is reported. Figure 11 shows representative images of each class.

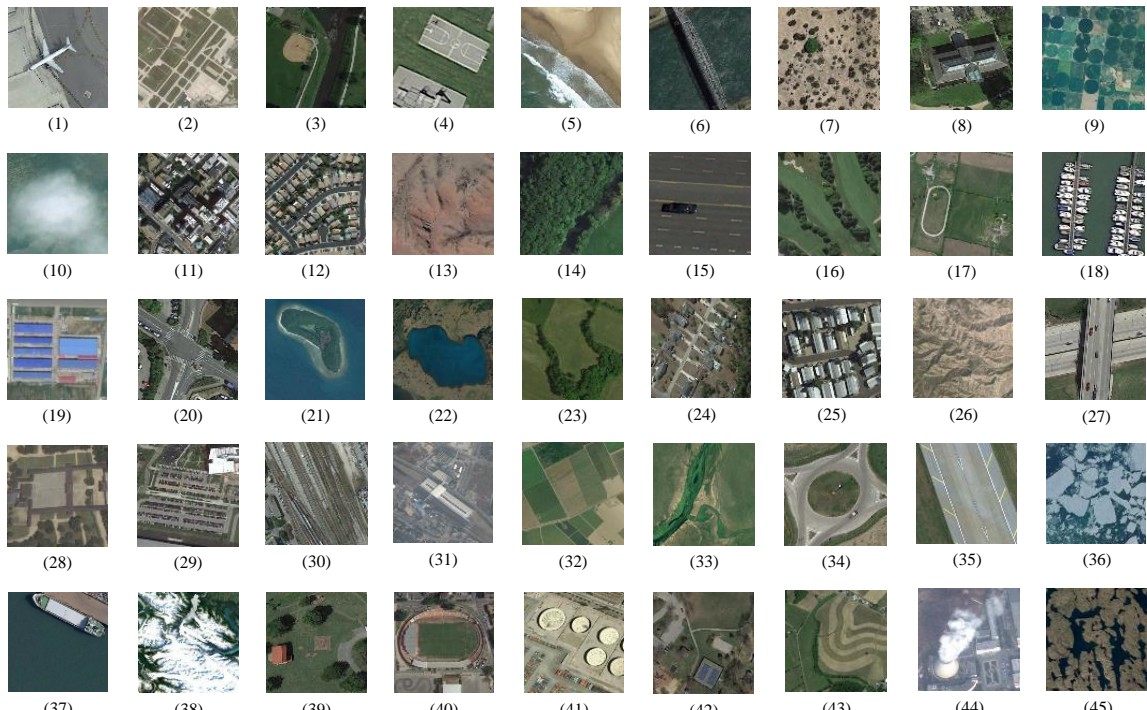

**Figure 11.** Representation of classes in the NWPU-RESISC45 dataset. (1) Airplane. (2) Airport. (3) Baseball diamond. (4) Basketball court. (5) Beach. (6) Bridge. (7) Chaparral. (8) Church. (9) Circular farmland. (10) Cloud. (11) Commercial area. (12) Dense residential. (13) Desert. (14) Forest. (15) Freeway. (16) Golf course. (17) Ground track field. (18) Harbor. (19) Industrial area. (20) Intersection. (21) Island. (22) Lake. (23) Meadow. (24) Medium residential. (25) Mobile home park. (26) Mountain. (27) Overpass. (28) Palace. (29) Parking lot. (30) Railway. (31) Railway station. (32) Rectangular farmland. (33) River. (34) Roundabout. (35) Runway. (36) Sea ice. (37) Ship. (38) Snowberg. (39) Sparse residential. (40) Stadium. (41) Storage tank. (42) Tennis court. (43) Terrace. (44) Thermal power station. (45) Wetland.

As shown in Table 2, the same as with the UC Merced dataset, the traditional methods based on handcrafted features, such as LLC and BoVW, perform significantly worse than deep learning methods. It can be seen in Figure 12, compared with VGG and GoogLeNet, our method also gives better classification results. The results show that introducing a multi-scale features extraction module to ResNet-18 increased the accuracy by 4.6%. It can be seen that the introduction of multiscale features

greatly improved the NWPU-RESISC45 dataset. The reason may be that the NWPU-RESISC45 dataset is a more complex dataset, and the multiscale feature extraction module can extract more information. After adding the attention mechanism, there was a 0.98% improvement in the proposed method. It can be clearly seen in the figure that the proposed method performs better than the other models listed. Compared with the other two datasets, the NWPU-RESISC45 dataset has a more complex structure. At the same time, the proposed method has the most obvious effect on the NWPU-RESISC45 dataset, which further illustrates that the proposed model can handle complex remote sensing images. Figure 13 shows the class accuracies of ResNet-18 and MSAA-Net on the NWPU-RESISC45 dataset, and MSAA-Net achieves the highest accuracy in 35 classes.

**Table 2.** Comparison between the previous reported accuracies with NWPU-RESISC45 dataset, with 50% training samples selected.

| Method | Classification Accuracy (%) |
| --- | --- |
| LLC | 59.92 |
| BoVW | 67.65 |
| AlexNet | 79.92 ± 2.1 |
| VGG-16 | 90.26 ± 0.74 |
| GoogLeNet | 91.45 ± 1.24 |
| GSoP-Net | 91.206 ± 1.32 |
| ECA-Net | 93.378 ± 0.26 |
| ResNet-18 | 89.93 ± 0.34 |
| MSAA-Net without attention | 94.03 ± 0.72 |
| **MSAA-Net (ours)** | **95.01 ± 0.54** |

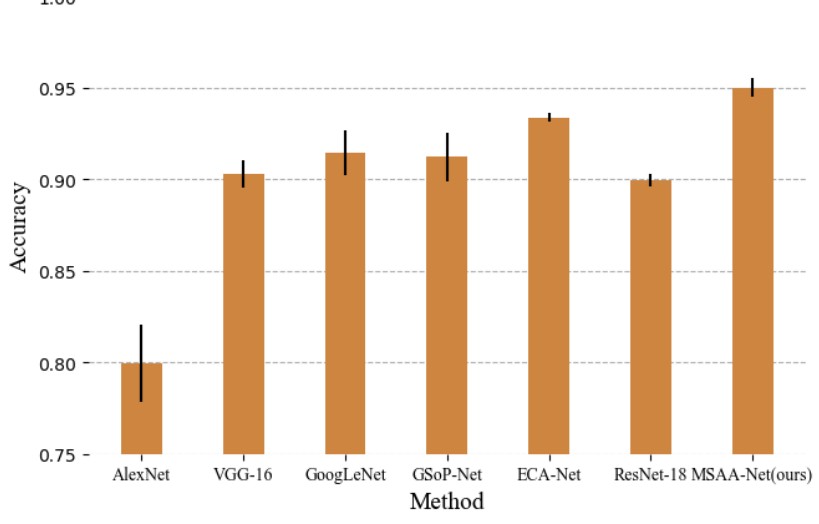

**Figure 12.** Classification results of different methods for NWPU-RESISC45 dataset.

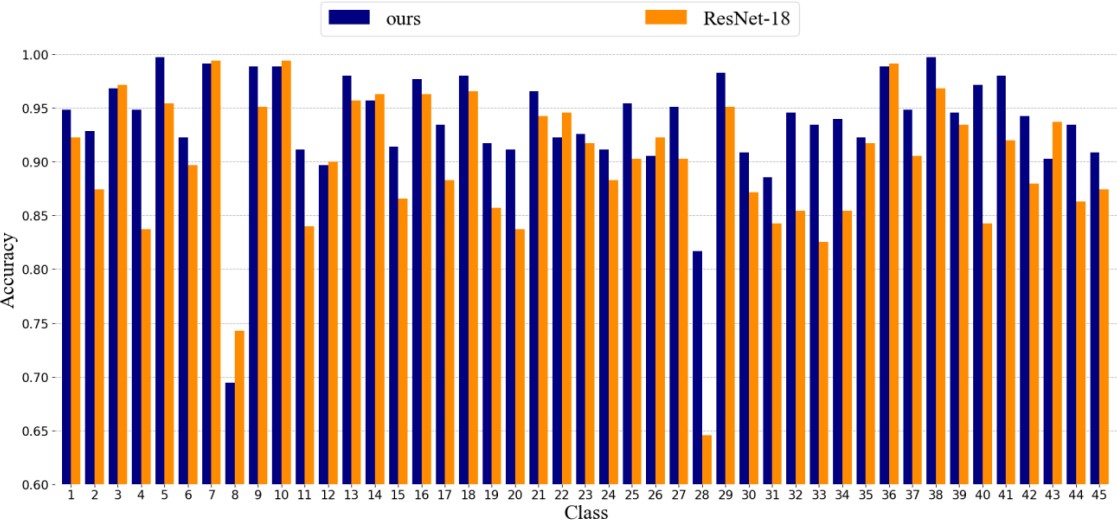

**Figure 13.** Accuracy in each class of ResNet-18 and MSAA-Net for the NWPU-RESISC45 dataset.

As shown in Figure 13, the classification results of our method in some classes are not better than ResNet-18, such as classes 8, 10, 26, and 43. The reason for this is that the proposed method is mainly for extracting high-level semantic information in the image, such as the objects. However, from the samples, it can be seen that the images in the four classes above mainly contain texture information. Therefore, in these classes, the proposed method does not show its advantages.

### 3.3. SIRI-WHU Dataset

The SIRI-WHU dataset was designed by the Intelligent Data Extraction and Analysis of Remote Sensing (RS_IDEA) Group at Wuhan University (SIRI-WHU), and contains 12 classes. There are 200 images in each category, and each image has a size of $200 \times 200$ pixels and a spatial resolution of 2 m per pixel. The same as the experiment on the UC Merced dataset, 80% of the dataset is used for training samples, and the rest is used for testing. Each experiment was repeated 10 times, and the average classification accuracy was reported. The 12 classes are agriculture, commercial, harbor, idle land, industrial, meadow, overpass, park, pond, residential, river, and water. Figure 14 shows an image from each class.

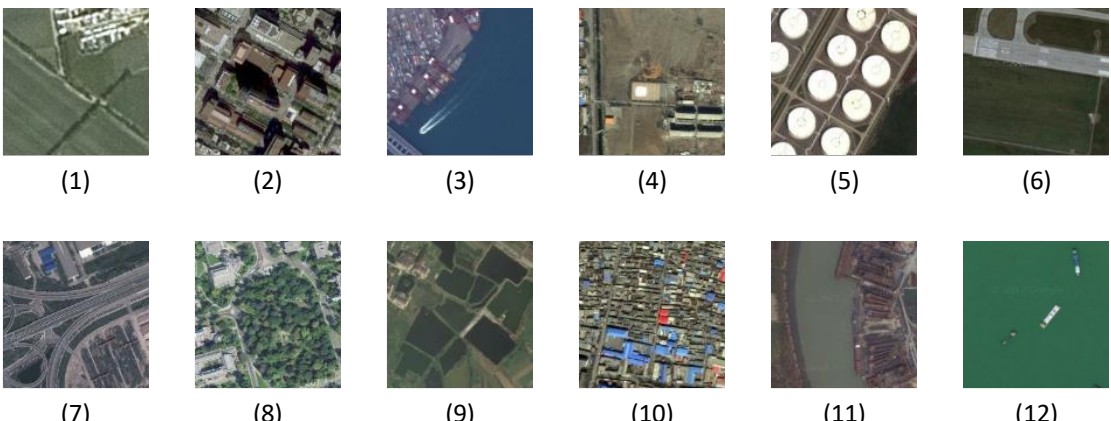

**Figure 14.** Representation of classes in the SIRI-WHU dataset. (1) Agriculture. (2) Commercial. (3) Harbor. (4) Idle land. (5) Industrial. (6) Meadow. (7) Overpass. (8) Park. (9) Pond. (10) Residential. (11) River. (12) Water.

As shown in Table 3, for the Google dataset of SRI-WHU, the proposed network performs better than the others. Compared with traditional methods that use handcrafted features for classification,

such as LLC and BoVW, the accuracy of the proposed method has improved a lot, by more than 10%. It can be seen that, compared with classic networks, such as VGG and GoogLeNet, our method performs better—by 4.4% and 2.9%, respectively. Compared with the excellent models proposed last year (GSoP-Net and ECA-Net), the proposed method also has slightly better performance. It is worth noting that, compared with ResNet-18, the introduction of the multiscale method only leads to a 1.72% better performance and the introduction of the attention mechanism increased performance by 1.26% for the Google dataset of SRI-WHU. The reason for this may be that the Google dataset of SRI-WHU is too simple, and there is no rich multiscale information in the dataset. As shown in Figure 15, the proposed method, again, achieved the best performance for the Google dataset of SIRI-WHU. Figure 16 shows the class accuracies of ResNet-18 and MSAA-Net on the Google dataset of SIRI-WHU, and MSAA-Net performs better than ResNet-18 in six classes and achieves the same results as ResNet-18 in four classes. It can be seen that the classification results of our method are not better than ResNet-18 in classes 2 and 3. The reason for this is that the number of samples in the Google dataset of SIRI-WHU is small and the image content is relatively simple. It can be seen from the samples that the objects in the images in these two categories are relatively small and the size of the objects hardly change, therefore the larger receptive field in the proposed method does not lend advantages.

**Table 3.** Comparison between the previously reported accuracies for the Google dataset of SIRI-WHU, with 80% training samples selected.

| Method | Classification Accuracy (%) |
|---|---|
| BoVW | 73.93 |
| LLC | 70.89 |
| AlexNet | 87.27 ± 1.63 |
| GoogLeNet | 92.31 ± 1.64 |
| VGG-16 | 90.83 ± 1.9 |
| GSoP-Net | 94.37 ± 0.56 |
| ECA-Net | 93.52 ± 0.4 |
| ResNet-18 | 92.23 ± 0.9 |
| MSAA-Net without attention | 93.958 ± 1.12 |
| MSAA-Net (ours) | **95.21 ± 0.65** |

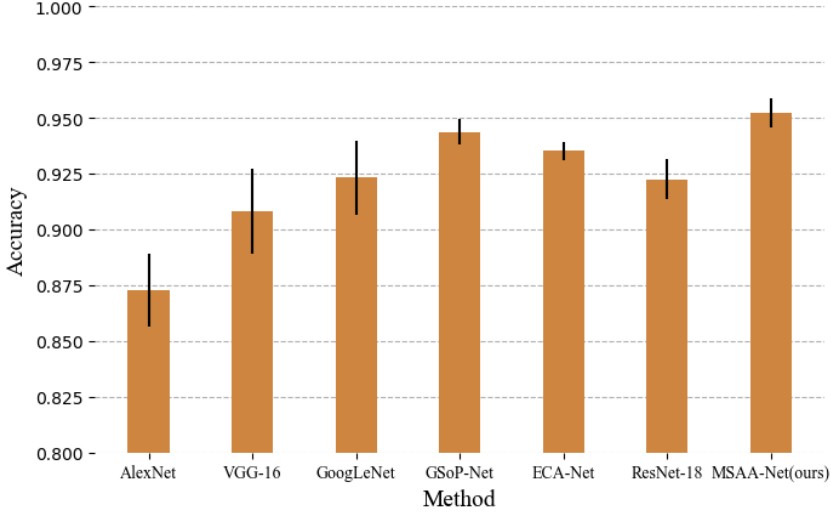

**Figure 15.** Classification results of different methods for the Google dataset of SIRI-WHU.

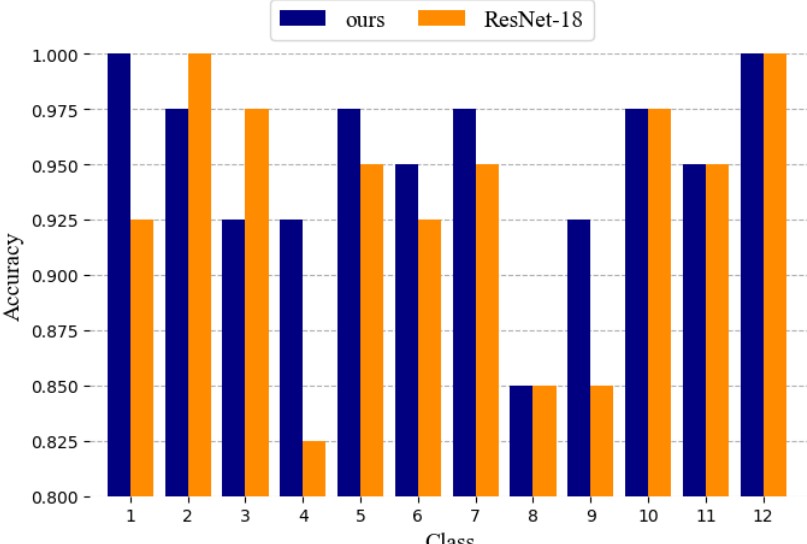

**Figure 16.** Accuracy in each class of ResNet-18 and our method for the Google dataset of SIRI-WHU.

## 4. Discussion

To explore the contribution of the multifeature extraction module and feature-fusion module, the feature-fusion method in MSAA-Net is compared with other fusion methods in this section. Then, the dilated convolution is used to expand the receptive field, and the dilation rate in the multiscale features extraction module is discussed. In addition, the depth of the multiscale extraction module is also analyzed. Furthermore, a comparison of the proposed method and the existing models in terms of training time is also given in this section. Finally, to understand how self-adaptive selection works, we analyze the attention weights in the proposed fusion module by inputting the same target object but in different scales in this section.

### 4.1. Fusion Method

To illustrate that the proposed fusion method is more effective than others, other fusion methods, such as adding element-wise and concatenation, are compared with the proposed method. The accuracy of MSAA-Net on the SIRI WHU dataset with different fusion methods in two feature-fusion modules is shown in Table 4. It can be seen that combining multiscale features has a good influence on the classification results. Whether directly adding features of different scales or concatenating by channel, the two feature maps are considered to contribute equally to the classification results, which does not make for flexible use of the information in multiscale features. As shown in Table 4, the proposed adaptive fusion method performs better than the other two simple fusion methods, by 0.63% and 0.42%. The reason is that a set of weights is learned from the global information to balance the proportion of features of different scales when fused, so that the model can adaptively and flexibly select features that are beneficial to the classification results. It is worth noting that the method of concatenation by channel is also slightly better than the method of adding element-wise. However, method concatenation by channel will double the number of output feature map channels, which will introduce additional parameters in the next convolutional layer. The proposed adaptive fusion method introduces a few additional parameters.

**Table 4.** Comparison between different fusion methods on the Google dataset of SIRI-WHU.

| Fusion Method | Adaptive Fusion (Ours) | Eltsum | Concat |
|---|---|---|---|
| **Classification Accuracy (%)** | **95.21** | 94.58 | 94.79 |

### 4.2. Dilation Rate

In the proposed method, dilated convolution instead of general convolution is used to expand the receptive field of the convolutional kernel, and the dilation rate determines the size of the receptive field. The accuracy of MSAA-Net on the SIRI WHU dataset with different dilation rates pair in two multiscale features extraction branches is shown in Table 5. It can be seen that the dilation rate in the two-branches setting (1,2) is better than others. The reason for this is that when the dilation rate increases, the network intends to capture the information with a large scale, but ignores the details. The general convolution $3 \times 3$ which means the dilation rate is 1, is a classic set in many deep learning methods, so it is important to have a dilation rate of 1 in one branch. Compared with a big dilation rate, setting the dilation rate in the other branch to 2 can balance the receptive fields and the detailed information.

**Table 5.** Accuracy on the Google dataset of SIRI-WHU with different dilation rates in the multiscale feature extraction module.

| Dilation rate in two branches | 1, 2 | 1, 3 | 2, 3 |
|---|---|---|---|
| Classification accuracy (%) | **95.21** | 94.58 | 94.14 |

### 4.3. Depth of Multiscale Extraction Module

In order to set a reasonable depth for the multiscale feature extraction module, an experiment on the multiscale feature extraction module on the SIRI WHU dataset was performed. The same as ResNet-18, the proposed model has eight blocks that contain two convolutional layers, so the block number in the multiscale extraction module is taken as a parameter in the experiment. The accuracy of the SIRI WHU dataset is shown in Table 6. It can be seen that the increase in the depth of the multiscale feature extraction module does not bring about an improvement; in fact, the classification effect is worse because the depth of the classification module is reduced. The reason for this is that the features about the size of the object in the image are low-level features, so it is not necessary to use many convolutional layers. On the contrary, too many convolutional layers will introduce a lot of parameters, which will make the model difficult to converge, so that it easily falls into overfitting problems.

**Table 6.** The accuracy of the Google dataset of SIRI-WHU with different convolution block numbers in the multiscale feature extraction module.

| Block Number in Multiscale Features Module | 1 | 2 | 3 |
|---|---|---|---|
| Classification Accuracy (%) | **95.21** | 94.375 | 94.164 |

### 4.4. Computational Time

The exact time comparison among the proposed method and some classic deep learning methods on the SIRI-WHU dataset is shown in Table 7. Each experiment was repeated 10 times, and the average training time was recorded. Due to it containing the simplest structure, AlexNet has the shortest training time, but its accuracy is also the lowest. It can be seen that, due to the introduction of a parallel structure, the training speed of the proposed method will be slower than ResNet-18. ECA-Net introduces an ECA layer in each residual block and is also based on the structure of ResNet-18, so its training time is also slightly shorter than that of the proposed model. Due to the introduction of a parallel multiscale feature extraction module, the backpropagation (BP) algorithm requires more calculations during the training process. As discussed in the previous subsection, the block number of multiscale features module in the proposed model is set to 1, so the training time only becomes a little longer than that of ResNet-18. In addition, compared with other classic models in the table, the proposed model has the shortest training time and the highest accuracy.

**Table 7.** The training time and accuracy on the Google dataset of SIRI-WHU.

| Method | Time (Each Epoch) (s) | Time (Total) (s) | Classification Accuracy (%) |
|---|---|---|---|
| AlexNet | 10 | 750 | 87.27 |
| VGG-16 | 23 | 1909 | 90.83 |
| GoogLeNet | 16 | 1088 | 92.31 |
| ResNet-18 | 10 | 790 | 92.23 |
| GSoP-Net | 17.6 | 1672 | 94.37 |
| ECA-Net | 10.5 | 840 | 93.52 |
| MSAA-Net(ours) | 11 | 1050 | 95.21 |

### 4.5. Self-Adaptive Selection

To understand how self-adaptive selection works, ten images randomly selected from the UC Merced dataset were cropped into different scales. Then these images were resized to $256 \times 256$ pixels and fed into MSAA-Net to obtain different attention values in the proposed fusion module. As shown in Figure 17, the smaller the image was cropped, the larger the target in the resized image became. As described in Section 2.3, in the fusion module, the attention weights for the two inputs were $p$ and $q$. $p$ represents the importance of the features extracted by the convolutional layer of the small receptive field and $q$ indicates the large one. $p - q$ is calculated as attention difference (attention value of feature with small receptive field minus that with large receptive field). As shown in Table 8, when the target object is larger, the mean attention difference decreases, which suggests that the fusion module will adaptively select more information from the feature map with a large receptive field. This shows that the proposed method can adaptively select features with a receptive field of an appropriate size when dealing with different sizes of targets.

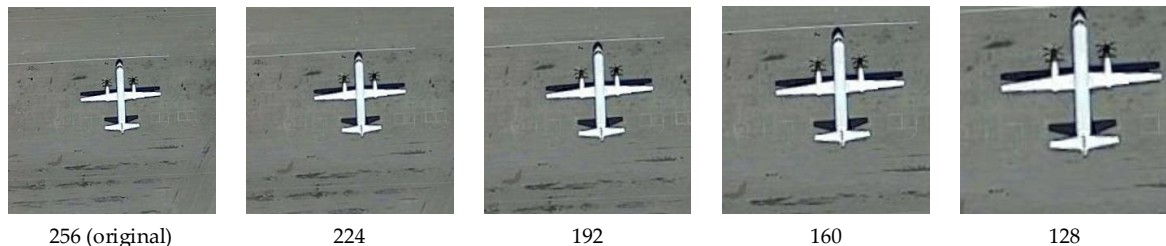

| 256 (original) | 224 | 192 | 160 | 128 |

**Figure 17.** Example of cropped and resized images.

**Table 8.** The mean attention difference in different scales on the UC Merced dataset.

| Scale of Cropped Samples (Pixels) | 256 (original) | 224 | 192 | 160 | 128 |
|---|---|---|---|---|---|
| Mean Attention Difference | 0.065 | 0.063 | 0.06 | 0.057 | 0.054 |

## 5. Conclusions

To reduce the influence of the scale variation of the objects in an image-on-scene classification, this paper presents a novel model named MSAA-Net. In the MSAA-Net, two sets of convolutional blocks of different receptive field sizes are used to capture objects of different sizes, which makes full use of the flexibility of convolution operations and does not introduce additional parameters. Inspired by the adaptive advantages of the attention mechanism, we propose a multiscale feature-fusion module, which can use the global information of the feature map to adaptively select useful information from two input features for fusion. In addition, the residual block and attention operation are adopted for accurate scene classification.

Experiments show that the features extracted by MSAA-Net are robust and effective compared with state-of-the-art methods for remote sensing images. In the proposed method, multiscale features are extracted from the shallow layer and penetrate into the deep network layer by layer. In future work, we will research how to extract effective multiscale features directly in deep convolutions.

**Author Contributions:** Conceptualization, L.L. and P.L.; methodology, P.L.; software, P.L.; validation, P.L., L.L. and X.G.; formal analysis, P.L.; investigation, C.S.; resources, J.M.; data curation, P.L.; writing—original draft preparation, P.L.; writing—review and editing, L.L.; visualization, J.M.; supervision, X.G.; project administration, L.J.; funding acquisition, F.L. All authors have read and agreed to the published version of the manuscript.

**Funding:** This research was funded by the State Key Program of National Natural Science of China: 61836009, Project supported the Foundation for Innovative Research Groups of the National Natural Science Foundation of China: 61621005, the Major Research Plan of the National Natural Science Foundation of China: 91438201, 91438103, the National Natural Science Foundation of China: U1701267, 61772399, U170126, 61906150, 61773304, 61902298 and 61801124, the National Science Basic Research Plan in Shaanxi Province of China: 2019JQ659, the New Think-tank of Department of Education In Shaanxi Province of China:20JT021, 20JT022, 20JY023, Science and Technology Program of Guangzhou, China: 201904010210.

**Acknowledgments:** The authors would like to thank the Assistant Editor who handled our paper and the anonymous reviewers for providing truly outstanding comments and suggestions that significantly helped us improve the technical quality and presentation of our paper.

**Conflicts of Interest:** The authors declare no conflict of interest.

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
