# Peer review of "A Multiscale Self-Adaptive Attention Network for Remote Sensing Scene Classification"

_remotesensing, doi:10.3390/rs12142209_

Round 1
Reviewer 1 Report
The authors have answered all m
The authors have answered all my comments. No further questions.
y comments. No further questions.
Reviewer 2 Report
Thanks for this work.
This is a resubmission paper. The authors did improve the quality of presentation, but I still feel the concept is not convincing to me.
- "Multi-scale" - You combine two convolutional blocks and fuse them, which is not really multi-scale features. Multi-scale is about input patch size and fusing multiple CNNs.
- I don't think this method is self-adaptive to multiple scales, unless you provide some concrete evidences.
- The results look interesting and the benchmark comparison is sufficient, but the details is not discussed. For example, in Figure 13, the classes 8 and 43 ResNet 18 is better than your method. The authors did not explain this phenomenon.
Reviewer 3 Report
Dear Author(s),
Thank you for your efforts.
The article still has many grammatical or spelling errors, making it not suitable for publication in the current form.
Regards.
Round 2
Reviewer 2 Report
Thank you for the work. I now accept for publication. Thanks!
Reviewer 3 Report
The manuscript has been enhanced to an acceptable level that it could be published.
This manuscript is a resubmission of an earlier submission. The following is a list of the peer review reports and author responses from that submission.
Round 1
Reviewer 1 Report
-This paper aims to extract semantic information and robust information for remote sensing scene classification. Besides, the paper proposed a self-adaptive attention network (MSAA-Net) that is tested and verified using three widely-used remote sensing datasets.
-The abstract is contained many grammatical errors and did not present the contribution of the paper. Enhance the abstract to focus only on the objectives, methodology, and add quantitative results.
-Conclusions should be rewritten so that it summarizing the main findings and empathizing on their usefulness for the scientific community.
The research paper should be written from the perspective of the third person. Words such as "we", "our" as mentioned in L19, L20, L21, L126, L399, L406, L430, L447, etc., which needs to be avoided and revised.
-The problem formulation is not well-defined, and further details of the proposed method are required. Therefore, the paper needs to be precisely revised.
-It is not clear if the work proposed is a new method or implementing some previous technique. Therefore, you need to add one paragraph about how the proposed method is deployed.
-A strong argument on the contribution and the validation of results should be provided.
-Rearrange the related work that you presented in the introduction section to be in levels based on the method used. Moreover, critically analyses the finding of them so that you can compare the proposed work with them.
- Needs more comprehensive evaluations and comparisons with other researchers for validating the obtained results supported by graphical and tabular data.
- Need to enhance the figures' resolutions to be more readable, add the measurement scale and Axis titled and units.
In lines 214-215:
The key point of this part is selecting effective information adaptively in two input features which differs in scale. By global pooling, global information is obtained ….
It is not clear how you could select the information for the different scale shapes. Also, how you obtained global pooling and global information. Add clarification.
The language used should adequately inform the reader, and Proofreading is mandatory for English grammar and style. See the following examples:
L14: different object size in …. should be ….. different object sizes in
L15: multiscale…. should be ….. a multi-scale ( Unify it in all text).
L17: module and deep …. should be ….. module, and a deep
L17: which contains features …. should be ….. that contains features
L17-18: In multi-scale feature extraction …. should be ….. In the multi-scale feature extraction
L18: with different receptive field …. should be ….. with different receptive fields
L19: features in different size …. should be ….. features in a different size
L19: and in feature fusion module …. should be ….. and in the feature fusion module
L19: squeeze-and-excitation …. should be ….. a squeeze-and-excitation
L14: different object size in …. should be ….. different object sizes in
L14: different object size in …. should be ….. different object sizes in
L14: different object size in …. should be ….. different object sizes in
L23 : and the results demonstrate the MSAA-Net are effective. Clarify in term to what factors the proposed method is effective.
L214: features which differs in scale …. should be ….. features, which differ in scale
Reviewer 2 Report
Dear authors,
Thanks for this work.
There are many parts I need your detailed clarification. It is not very clear to me where is the multi-scale in your method. Did you use different window sizes of convolutional input feature map? How did you decide the scale parameter?
Why do you use spatial attention in your network? What happens if you don't have spatial attention layer?
Also, why the Residual Networks used 18 layers? In my mind, the common Residual Nets would have 50 layers.
The main conclusion is to increase classification accuracy. What about computational efficiency? How long is your method taking compared with other benchmarks?
Reviewer 3 Report
1. A number of similar recent articles have not been considered, for example:
- Geospatial Object Detection via Deconvolutional Region Proposal Network https://ieeexplore.ieee.org/document/8735963
- Ground and Multi-Class Classification of Airborne Laser Scanner Point Clouds Using Fully Convolutional Networks https://www.mdpi.com/2072-4292/10/11/1723
Compare “Figure 5. The adaptability of DeRPN to object rotation and scale variance” in this article and “Figure 1. Images of the same category in which objects have different size. (a) airplane class. ” in a peer-reviewed article. It is almost the same thing. By the way, there is no link to Figure 1.
- Object Detection with Low Capacity GPU Systems Using Improved Faster R-CNN https://www.mdpi.com/2076-3417/10/1/83
2. Compared with the known methods, the authors demonstrate an improvement of 0.42% to 0.63% (Comparison between different fusion methods) and 0.833% to 1.458% (Classification accuracy).
I understand why the authors did not indicate these numbers in the abstract. Such a minor improvement will immediately discourage readers from wasting their time reading this article. However, it will be honest for authors to point out this slight improvement right in the abstract for the respect of readers.
3. It is unclear how, having only two feature maps, you can talk about multiscale information.
Round 2
Reviewer 1 Report
Thank you for your efforts.
The article still has many grammatical or spelling errors that make the meaning unclear and sentence construction errors, punctuation errors. I sent the editor some examples.
I will consider this as a major correction.
Reviewer 2 Report
The answers provided by the authors are not clear to me.
- Where is "multi-scale" in your proposed method?
- Although you cite Woo et al. [49], you need to compare with and without spatial attention layer in your datasets.
- So your method for ResNet-50 is not good. That's fine. You need to further apply VGG-16 to demonstrate robustness.
- I need to see exact computational time comparison among different methods.
